# The Efficiency Improvement of Multiple Receivers in Wireless Power Transmission by Integrating Metasurfaces

**DOI:** 10.3390/ma15196943

**Published:** 2022-10-06

**Authors:** Jian-Hui Xun, Yajie Mu, Kunyi Zhang, Haixia Liu, Long Li

**Affiliations:** 1School of Cyber Science and Engineering, Qufu Normal University, Qufu 273165, China; 2Key Laboratory of High Speed Circuit Design and EMC of Ministry of Education, School of Electronic Engineering, Xidian University, Xi’an 710071, China

**Keywords:** magnetic coupled resonators, wireless power transfer, multiple receivers, negative permeability metasurface, transfer efficiency improvement

## Abstract

In this paper, we propose the use of metasurfaces to enhance evanescent wave coupling to improve the wireless power transfer (WPT) efficiency of multiple receivers. A 4 × 4 negative permeability metasurface is designed and placed between the transmitter (Tx) and receiver (Rx) coils for the greatest improvement in transfer efficiency. Through the analysis of the number and position topologies of Rx coils, the efficiency can be greatly improved; the maximum efficiency at longer transmission distances is achieved through the 4 × 4 negative permeability metasurface in the multiple−receiver system. We show with simulation and measurement results that the power transfer efficiency of the system can be improved significantly by integrating metasurfaces. The maximum transfer efficiency is achieved in a multiple−receiver WPT system when the number and topology of Rx coils is case 0 of single transmitter−three receivers (STTR). The results show that the total efficiency of the multiple receivers WPT system can be as high as 97%.

## 1. Introduction

Wireless energy transfer (WPT) technology realizes space wireless energy transfer through electric field coupling, magnetic field coupling, and electromagnetic radiation. This eliminates the limitations of wired charging, which has aroused the extensive attention of numerous scholars at home and abroad on WPT. Among them, magnetic resonance coupling wireless power transfer technology can achieve medium− and long−distance, high−power, and high−efficiency energy transmission [1,2,3]. It is one of the most widely studied and applied technologies in many wireless energy transmission modes. With the rapid development of WPT technology, how to further improve the efficiency of magnetic coupling resonant wireless energy transmission is very important. While improving the efficiency of magnetic coupling resonant WPT, how to improve the efficiency of multiple receivers WPT is particularly urgent with increasingly diverse demands. The main purpose of this paper is to integrate metasurfaces technology based on the existing magnetic resonant coupled wireless energy transmission. Thus, the efficiency and transmission distance of magnetic coupled resonant WPT can be further improved.

When the frequency of the excitation source in the circuit is equal to the natural frequency of the circuit, the electromagnetic oscillation in the circuit will reach its peak. Magnetic resonance WPT is based on this principle. Two Tx and Rx coils with the same resonant frequency are used to generate high−intensity electromagnetic coupling at a certain distance to carry out wireless energy transmission [4,5,6]. In 2007, the Massachusetts Institute of Technology (MIT) lit a 60W bulb at a distance of 2 m by magnetic resonance coupling WPT technology. The transmission efficiency is from 40–50%. In 2008, Intel’s Seattle lab tested the technology several times, achieving 75% efficiency at a distance of 1 m. In 2010, Fujitsu of Japan first proposed the multi−load magnetic resonance wireless charging technology and realized the use of this technology to charge multiple devices at the same time. This is the beginning of the development of multi−device charging.

Magnetic resonance wireless energy transmission to multiple receivers has been attracting more attention due to the increasing demand for charging electronic devices [7,8,9,10,11,12,13]. The key issue of multiple receivers WPT systems is frequency splitting which is proposed in [8]. The frequency splitting occurs when two receivers are in close enough proximity that their magnetic fields are relatively strongly coupled. This makes the multiple−receiver WPT system has worse total efficiency than that of the single−receiver WPT system. In [9], two Rx coils are symmetrically located on both sides of the Tx coil to reduce the mutual coupling of the two Rx coils. Compared with a single Rx coil, the efficiency is improved. In [11], the power transmitted by an omnidirectional transmitter composed of three orthogonal circular coils will be received by any receiver located nearby. Due to the randomness of the receiving location, the coupling between the Rx coils is small, and the efficiency of the multiple receivers WPT system is improved. Therefore, the multiple receiving coils of the multiple receivers WPT with the best transmission efficiency at present are not on the same receiving surface. How to reduce the coupling between co−planar receiving coils is a crucial issue to improve the efficiency of multi-receiving WPT systems.

Metamaterials or metasurfaces are widely used to improve the WPT efficiency of their unique electromagnetic properties [14,15,16], which is not seen in natural materials, such as the zero refraction metamaterials, the negative refractive index (NRI) and negative permeability (MNG) metamaterials [17,18,19,20,21,22]. The authors in [14] propose two metamaterial slabs of a 3 × 3 array to enhance the magnetic coupling. The influence of electromagnetic metamaterials on the WPT system under the conditions of lateral misalignment and angular offset was investigated in [15]. Using a properly designed metamaterial slab between two coils, the coupling between coils can be enhanced, and the power transfer efficiency and the distance of efficient power transfer are significantly increased [16]. In [18], the metamaterials are used to enhance the evanescent wave coupling and improve the transfer efficiency of a WPT system based on coupled resonators. The transfer efficiency of a WPT system can be improved from 17% to 47% by the metamaterial, while there is only one Rx coil which is not suitable for multiple receivers. In [23], an improvement of 20% WPT efficiency for each Rx coil can be obtained by using a zero−refraction metamaterial, while the transfer distance is only 5 cm, which is not suitable for the longer transfer distance system. Moreover, most of the studies only integrate the metamaterials into the single transmitter and single receiver wireless power transfer (STSR WPT) system, which cannot satisfy the increasing demand for multiple electronic charging devices.

In this paper, a 4 × 4 negative permeability metasurface is designed to improve the efficiency of the multiple receivers WPT system. The simulation and measurement results show that the transfer efficiency of the system can be improved significantly by the metasurface. Then, the number and topologies of Rx coils in multiple receivers WPT system integrated with the metasurface are analyzed to achieve the maximum transfer efficiency. The simulation results show that the total efficiency of the multiple receivers WPT system can be as high as 97%. The simulation and measurement results of the total efficiency are 92% and 77.6% in a single transmitter−four receivers WPT system, respectively. In addition, the size of the Tx coil and the Rx coils, as well as the transfer distance, are all suitable for conformal with the desk for convenient charging.

The innovations of this paper are as follows:(a)Using a 4 × 4 negative permeability metasurface placed between Tx and Rx coils of the multiple receivers WTP system, the coupling between Tx and Rx coils can be enhanced and the power transfer efficiency of WPT is significantly increased.(b)Through the analysis of the number and position topologies of Rx coils, the mutual of Rx coils is reduced, which can eliminate the coupled mode frequency splitting in the multiple receivers WPT system. The efficiency of multi−receiving WPT systems is significantly improved.

This paper is organized as follows. In Section 2, the principle of negative permeability metasurface in WPT for improved efficiency is analyzed. In Section 3, a negative permeability metasurface is designed for improving the conventional WPT system efficiency. The optimized position of the designed metasurface is analyzed. In addition, the number and topologies of Rx coils in multiple receivers WPT system integrated with the metasurface are analyzed. In Section 4, the measured results are given to verify that the negative permeability metasurface for the efficiency of multiple receivers WPT system has a significant improvement.

## 2. Principle of Negative Permeability Metasurface in WPT

Since the magnetic resonance coupling WPT is essentially the coupling of the evanescent wave, which can be used to improve the efficiency of WPT, tt has been proposed and shown with numerical simulation results that a negative permeability metasurface can be used to enhance the evanescent nearfield and eventually improve the transfer efficiency in WPT [24].

To analyze the evanescent wave amplification principle of negative permeability metasurface, the theoretical model of the WPT system loaded with the negative permeability metasurface is shown in Figure 1. The metasurface is assumed to be infinite in the plane parallel to the *xoz* plane, and its normal direction is parallel to the axial direction of the Tx coil. The incident wave is assumed to be TE polarized. The effective flux between the Tx and Rx coils propagates along the *y*-axis. The propagation distance of electromagnetic waves is divided into three regions: *y* < 0, 0 < *y* < *L*, and *y* > *L*. Tangential electric field and tangential magnetic field components in different regions are expressed as:
(1)$$H_{x}=\frac{E_{0}}{\omega \mu_{x}}e^{-jk_{x}x}\{\begin{array}{c} k_{y}\left(e^{-jk_{y}y}-Re^{jk_{y}y}\right),\;y<0\\ q_{y}\left(Ae^{-jq_{y}y}-Be^{jq_{y}y}\right),\;0\leq y<L\\ k_{y}Te^{-jk_{y}(y-L)},\;y>L \end{array}$$
(2)$$E_{x}=E_{0}e^{-jk_{x}x}\{\begin{array}{c} e^{-jk_{y}y}+Re^{jk_{y}y},\;y<0\\ Ae^{-jq_{y}y}+Be^{jq_{y}y},\;0\leq y\leq L\\ Te^{-jk_{y}(y-L)},\;y>L \end{array}$$

*E_0_* is the amplitude of the incident wave. *k_x_* and *k_y_* are wave numbers in the *x* and *y* directions, respectively. *k_0_* is the wave number in free space. *q_y_* is the wave number of the metasurface along the *y*-axis. *R* and *T* are reflection and transmission coefficients, respectively. *A* and *B* are the amplitudes of forwarding and reverse waves inside the metasurface, respectively. As for the continuity of tangential components of the electric field at the boundary of *y* = 0 and *y* = *L*, the transmission coefficient of the incident wave at the metasurface can be obtained as follows:(3)$$T=\frac{2q_{y}k_{y}}{{\left(q_{y}+k_{y}\right)}^{2}e^{-jq_{y}L}-{\left(q_{y}-k_{y}\right)}^{2}e^{jq_{y}L}}$$

For negative magnetic materials (ε = 1, μ < 0), *q_y_* can be written as:(4)$$q_{y}=\sqrt{\varepsilon \mu \left(k_{0}^{2}-k_{x}^{2}\right)}$$

It can be concluded from (4) that *q_y_* of the negative permeability metasurface is a purely imaginary number. When the equivalent permeability is close to −1, the transmission coefficient can be deduced as follows:
(5)$$\underset{\mu \to -1}{\lim}T=\underset{\mu \to -1}{\lim}\frac{2q_{y}k_{y}}{{\left(q_{y}+k_{y}\right)}^{2}e^{-jq,L}-{\left(q_{y}-k_{y}\right)}^{2}e^{jq_{y}L}}=\exp\left(-iq_{y}L\right)$$

It can be concluded from (5) that the exponential part of the transmission coefficient is greater than zero when the negative permeability metasurface is loaded. Therefore, the value of the transmission coefficient is greater than 1. The function of evanescent wave amplification is realized. Therefore, the negative permeability of the metasurface can form a focusing effect on electromagnetic waves. The electromagnetic wave that diverges exponentially with the propagation distance is converged into the Rx coil to improve the coupling of the Rx coil.

## 3. Multiple Receivers WPT Integrating with Metasurface

Next, we start to design and fabricate a negative permeability metasurface and a WPT system to prove the amplifying properties of the metasurface to evanescent waves.

Figure 2 shows the conventional multiple receivers WPT system. Note that the Tx and all the Rx coils have the same structure but different sizes. Both of them consist of an exciting coil and a 3−turn coupled coil. A little rectangular ground (GND) is reserved for welding the feedline and the π shape matching circuit, as shown in Figure 2a. The Tx coil is printed on the upper surface of F4B dielectric substrate with a thickness of 1 mm, its relative permittivity is 2.65, and the loss tangent is 0.001. The FRB dielectric has a little loss of electromagnetic waves that will not affect the transfer efficiency of the WPT system. The Rx coils are printed on the F4B dielectric substrate with a thickness of 1mm. The detailed geometrical parameters of the Tx coil are: s(Tx) = 45 cm, so(Tx) = 40.4 cm, se(Tx) = 17.6 cm, w(Tx) = 0.8 cm, sl(Tx) = 0.6 cm, g(Tx) = 2.5 cm. The geometrical parameters of the Tx and Rx coils are shown in Table 1. Figure 2b shows a perspective view of a conventional multiple receives WPT system. The distance of the adjacent Rx coils is labeled as d. The h is labeled as the transfer distance which is 80 cm. A π shape matching circuit was added to the Tx and Rx coils respectively to match at 13.5 MHz and obtain the maximum efficiency. The π shape matching circuit is analyzed and optimized in the Advanced Designed System (ADS). Ansys High−Frequency Structure Simulator (HFSS) was used to simulate the Tx and Rx coils.

Then, the efficiency of the conventional multiple−receiver WPT system was analyzed at different distances from the adjacent Rx coils. The efficiency of four Rx coils in multiple receivers WPT system can be calculated as [8]:(6)$$\eta =\frac{{\left|S_{21}\right|}^{2}+{\left|S_{31}\right|}^{2}+{\left|S_{41}\right|}^{2}+{\left|S_{51}\right|}^{2}}{1-{\left|S_{11}\right|}^{2}}$$

The S−parameters in (6) are calculated in the ADS. The Tx coil is port 1 and the Rx coils are ports 2, 3, 4, and 5. The S−parameter is used to reflect the relationship between the incident voltage wave and the reflected voltage wave at the port. *S*_11_ is the reflection coefficient of port 1 when the other ports are matched. *S*_21_, *S*_31_, *S*_41,_ and *S*_51_ are the transmission coefficients from port 1 to ports 2, 3, 4, and 5 when ports 2, 3, 4, and 5 are matched. Due to the perfect matching of the matching circuit, the value of ${\left|S_{11}\right|}^{2}$ at the operating frequency is nearly zero. Therefore, the efficiency can be calculated as
(7)$$\eta ={\left|S_{21}\right|}^{2}+{\left|S_{31}\right|}^{2}+{\left|S_{41}\right|}^{2}+{\left|S_{51}\right|}^{2}$$

Then, the efficiency of conventional multiple receivers WPT system is calculated and shown in Table 2. It can be seen from Table 2 that the total efficiency of the multiple receivers is extremely low where the max efficiency of Rx coils is only 0.31%. Compared with the efficiency of a single−receiver WPT system, the conventional multiple receivers WPT cannot be applied to any charging system. It can be seen that the efficiency still cannot be improved to more than 31% under the different distances of d. This efficiency is very low. Therefore, this multiple receivers WPT system needs further improve the transfer efficiency.

In order to improve the transfer efficiency of the multiple−receiver WPT system, a negative permeability metasurface is designed and placed between the Tx and Rx coils for improving transfer efficiency. Since the resonant frequency of the coil is very small, the wavelength of this frequency point is too long. This leads that the metasurface cell being over large. Considering that the metasurface cell needs to be compact in size, low in loss, and easy to fabricate, we choose a double−sided square spiral as the design of the cell for the negative permeability metasurface. Figure 3a shows the detailed geometry of the metasurface cell and its simulation model. The metasurface cell adopts the periodic boundary simulation method in HFSS (high−frequency simulation software). The two planes perpendicular to the z-axis are set as the magnetic boundary (PMC). The two faces perpendicular to the x-axis are set as electrical boundaries (PEC). The two planes perpendicular to the y-axis are set as wave ports, and the purpose of this setting is to let the magnetic field pass perpendicularly through the metasurface cell, as shown in Figure 3a. The two spirals of the metasurface cell are printed on the upper and lower layers of the 1 mm thickness F4B dielectric substrate. The upper and lower spiral structures are mirror symmetric about the center of the structure. The corresponding real and imaginary relative permeability is presented in Figure 3b. The relative permeability curve in Figure 3b is obtained by simulating the metasurface cell in HFSS and using the Smith inversion algorithm. It can be seen that the metasurface cell has a magnetic resonance of around 13.5 MHz where the relative permeability is negative. At 13.5 MHz, the relative permeability of the metasurface cell is around −1. Then, the negative permeability metasurface is composed of 4 × 4 metasurface cells with a total dimension of 68 cm × 68 cm to optimize the position of the Metasurface where the transfer efficiency is maximum. Increasing the number of the metasurface cell in WPT can increase the magnetic field strength at the Rx coils. The transfer efficiency of the system is improved. However, if the metasurface is too large, the magnetic field at the Rx coils will be weakened. Therefore, the number of the metasurface cells is generally 4 × 4, larger than that of the Tx coil in this paper.

As shown in Figure 4a, the proposed multiple−receiver WPT system is integrated with the desk. This is convenient for people that the charging device (Rx coil) is put on the surface of the desk, and the Tx coil is put on the ground. This desktop scenario does not influence the activity of the people. The distance between the metasurface and the surface of the four Rx coils is labeled as H. The metasurface can be integrated with the drawer of the desk. The model is simulated in HFSS to extract SNP files into ADS for port matching of Tx and Rx coils. The LC matching circuit is obtained, and then the S−parameters of the system are obtained by simulation and further optimization in HFSS, as shown in Figure 4b. Then, the efficiency of the multiple receivers WPT system integrated with metasurface is analyzed in different H. Table 3 shows the variation of WPT efficiency with the position of the metasurface. It can be seen that the efficiency has significant improvement after using the metasurface in the WPT system. The total efficiency can be as high as 92% when the metasurface is put 20 cm below the surface of the Rx coils. Compared with the conventical situation, the maximum efficiency of the multiple−receiver WPT system integrated with metasurface increases about 300 times. The S−parameters of the Tx coil and Rx coils in H = 20 cm are shown in Figure 4b. It can be seen that the 1 port of the Tx coil is matched at 13.1MHz, and the transmission coefficients from the Tx coil to the Rx coils are about −6 dB. In addition, the resonate frequency varies with the position of the metasurface. This is because the Tx and Rx coils influence the magnetic resonance of the metasurface. The optimal transfer efficiency has a frequency shift when H is different. This requires us to adjust the resonant frequencies of Tx and Rx coils according to the actual situation so that the maximum transmission efficiency after loading the metasurface is at the working frequency point. In addition, although the position of the metasurface at H = 20 cm is the best for improving the efficiency of the WPT system, there is still room for improving the efficiency of the multiple−receiver WPT system. That is, the position topology between Rx coils also has an impact on the total efficiency. Next, the influence of the position topologies between different Rx coils on the total efficiency is analyzed. In practical applications, the position and numbers of the charging devices may vary with the demands. Therefore, the efficiency of multiple receivers WPT in these situations should be analyzed. Next, we will present the optimal efficiency of the WPT system for the multiple receivers with different positions and different numbers when the position of the metasurface is at H = 20 cm.

### 3.1. Single Transmitter−Four Receivers (STFR)

The position topology of single transmitter−four receivers and the conventional topology is the same, which is evenly distributed. Figure 5 presents that the receiving ability of the four Rx coils varies with the distance of the adjacent Rx coils d. It can be seen that the highest efficiency is obtained when the distance of the adjacent Rx coils d is 50 mm. The whole tendency is that the total efficiency increases with d when d is less than the optimal distance. The efficiency decreases with the increase of d when d is greater than the optimal distance. A key issue for multiple receivers WPT is the coupled mode frequency splitting that occurs when two receivers are in close enough proximity that their magnetic fields are relatively strongly coupled [8]. This gives the multiple−receiver WPT system worse total efficiency when the mutual coupling of the Rx coils is strong. When the distance between the Rx coils is less than 50 mm, the coupling between the Rx coils is too strong, resulting in frequency splitting. As a result, the efficiency of the target frequency is reduced. When the distance between the Rx coils is greater than 50 mm, the coupling between the Rx coils becomes smaller, but the aperture efficiency becomes lower. Therefore, the transfer efficiency is also reduced. When the distance between the Rx coils is 50 mm, it is the optimal position for maximum efficiency after the coupling and aperture efficiency are balanced. Therefore, the four Rx coils should be kept 50 mm apart from each other in practical applications. In this situation, the efficiency of the third Rx coil is the largest, 30.25%. The lowest efficiency is 18.49%. This minimum efficiency is comparable with the efficiency of the single−transmitter–single−receiver WPT system. The total efficiency is 92.3%. The best transfer efficiency is achieved thanks to the introduction of the negative permeability metasurface into the multiple receivers WPT system.

### 3.2. Single Transmitter−Three Receivers (STTR)

Figure 6 shows the three possible arrangements for charging devices. The spacing distance of the adjacent Rx coils is labeled as d31, d32 as well as d33 in different cases. The total efficiency of the STTR WPT system in these three cases is presented in Figure 7. It can be seen that the optimized efficiency of the STTR WPT system can be as high as 97% in case 0 when d31 is 5 cm. The efficiency significantly decreases when d31 has a little change in case 0. The efficiency in case 1 can also be as high as 93% and the efficiency has a smaller decrease when d31 varies. Besides, the efficiency in case 2 is the lowest. Case 1 is the best topology way for the STTR WPT system. In addition, the efficiency of the STTR WPT system is 5% more than the STFR WPT system. Therefore, the position topology between different Rx coils has an important influence on efficiency. The STTR WPT system can obtain better efficiency than the STFR WPT system.

### 3.3. Single Transmitter−Two Receivers (STTWR)

Figure 8 shows the three possible arrangements for the two charging devices. The spacing distance of the adjacent Rx coils is labeled as d21 or d22. The total efficiency of the STTWR WPT system in these three cases is presented in Figure 9. It can be seen that the optimized total efficiency of the STTWR WPT system can be as high as 78% in case 3 when d21 is 5 cm. The total efficiency in case 4 can also be as high as 65%. Therefore, case 3 is the best topology way for the STTWR WPT system. Note that the total efficiency of the STFR WPT system and the STTR WPT system is higher than the STTWR WPT system. However, the receiving efficiency of the STTWR WPT system is more uniform than the STFR WPT system and the STTR WPT system.

### 3.4. Single−Transmitter–Single−Receiver (STSR) Systems

At last, we discuss the performance of the STSR WPT system for practical applications. Figure 10 shows the two kinds of arrangements ways for the single charging device. The distance between the Rx coil and the origin of the system is labeled as d11 or d12. The efficiency of the STSR WPT system in these two cases is presented in Figure 11. It can be seen that the optimized efficiency can be as high as 36% in both cases. Case 5 is the best topology way for the STSR WPT system whose efficiency is 38.7%.

Through the analysis of the number and position topologies of Rx coils above, it can be found that the efficiency reaches the maximum when the number of receiving coils is 3, which can reach 97%. In addition, it can be found that the efficiency is higher when there are more Rx coils, and the efficiency is minimum when there is only one Rx coil. Therefore, the negative permeability metasurface has the most obvious efficiency improvement for multiple receivers WPT systems. That is, it is most suitable for multiple receivers WPT systems.

## 4. Measurement Results

Figure 12 shows the measurement setup for STFR WPT system integrating with the fabricated negative permeability metasurface. The spacing between the Rx coils is 5 cm. The vertical distance from the Tx coil to the Rx coils is 80 cm, and the distance from the metasurface to the Rx coils is 20 cm. According to Section 3.1, under these parameter settings, SFFR WPT can obtain the maximum transfer efficiency. In order to simulate the actual application scenario, the metasurface and the Rx coils are installed on the desk, and the spacing between the two is simulated by a stack of books. The Tx coil is installed on the floor. The vector network analyzer is used to measure the S−parameters. The efficiency can be obtained by the S−parameters. The comparison of measured and simulated results of the transfer efficiency of the STFR WPT system integrating with metasurface is presented in Figure 13. Note that here d represents the d11 in the STSR WPT system. It can be seen that the simulation efficiency of the STFR WPT system is higher than the measurement results. The maximum efficiency of measurement is 77.6% when d is 5 cm, which is consistent with the simulation result.

The total efficiency of the proposed STFR WPT system and literature [8,18,23] are summarized in Table 4. Compared with the single−transmitter–single−receiver (STSR) system without metasurface in [8], the proposed STFR WPT system integrated with the negative permeability metasurface achieves a higher efficiency under a much longer transfer distance. Compared with the single−transmitter−single−receiver (STSR) system with metasurface in [18], the proposed STFR WPT system also has higher efficiency. It can be seen from [23] and the proposed STFR WPT that the multiple−receiver WPT system has higher efficiency than the single−transmitter−single−receiver (STSR) WPT system [18] when the metasurface is used. In addition, the STFR WPT system integrated with the negative permeability metasurface achieves higher efficiency than [23] under a much longer transfer distance. Therefore, this paper achieves the maximum transfer efficiency and transmission distance in the multiple−receiver WPT system.

## 5. Conclusions

A negative permeability metasurface is designed for improving the efficiency of mid−range wireless power transfer to multiple receiver systems. Using a 4 × 4 negative permeability metasurface placed between Tx and Rx coils of the multiple receivers WTP system, the coupling between Tx and Rx coils can be enhanced and the power transfer efficiency and the distance of efficient power transfer are significantly increased. Through the analysis of the number and position topologies of Rx coils, the mutual of Rx coils is reduced, which can eliminate the coupled mode frequency splitting in the multiple−receiver WPT system, significantly improving its efficiency. The results show that the power transfer efficiency of the system can be significantly improved by the metasurface. In addition, the single−transmitter−three−receiver (STTR) system is the optimal number and topology of receiver (Rx) coils in a multiple−receiver WPT system integrated with a metasurface. This system’s efficiency can reach 97%. The single−transmitter−four−receiver (STFR) WPT system integrated with the metasurface achieved 92% efficiency. The simulation and measurement results of the total efficiency were 92% and 77.6% for the single−transmitter–four−receiver (STFR) WPT system, respectively. Therefore, the negative permeability metasurface has the most obvious efficiency improvement for multiple−receiver WPT systems. That is, it is most suitable for multiple−receiver WPT systems. However, the space distance of Rx coils is too small, which limits the range of multi−target charging. Next, we need to further improve the distance between the Rx coils in the multiple−receiver system and maintain a high transfer efficiency. The application scenarios of the multiple−receiver WPT system integrating negative permeability metasurface designed in this paper are various desks, including home desks. The Tx coil can be mounted on the floor. The metasurface is installed under the desk drawer. The Rx coils are installed on the surface of the table to achieve multi−target charging.

## Figures and Tables

**Figure 1 materials-15-06943-f001:**
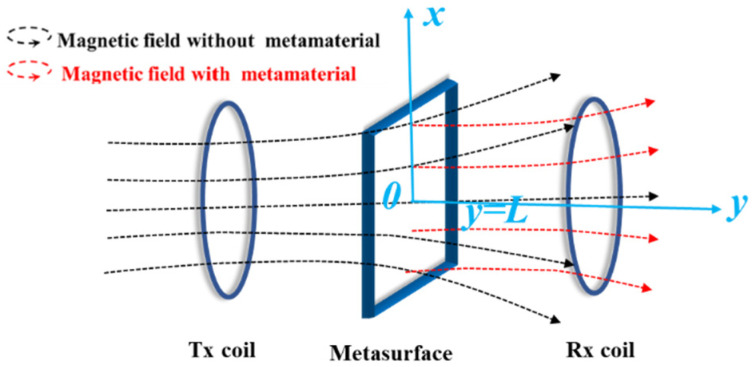
The WPT system for loading the negative permeability metasurface.

**Figure 2 materials-15-06943-f002:**
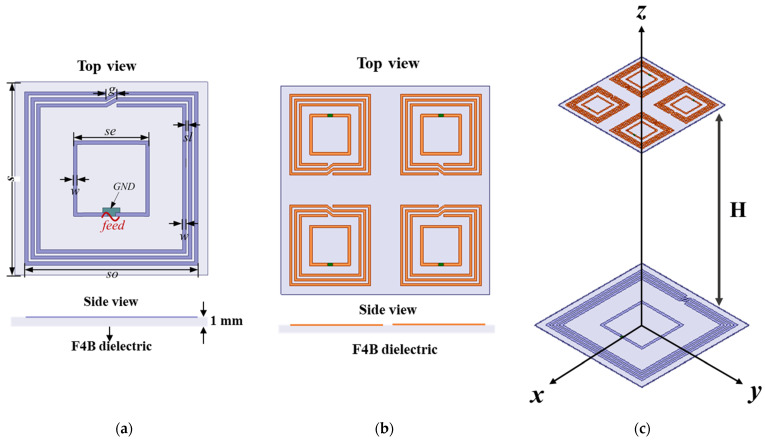
The Tx coil and multiple receivers WPT system. (**a**) Top view and side view of the Tx coil. (**b**) Top view and side view of the Rx coil. (**c**) The conventional multiple receivers WPT system.

**Figure 3 materials-15-06943-f003:**
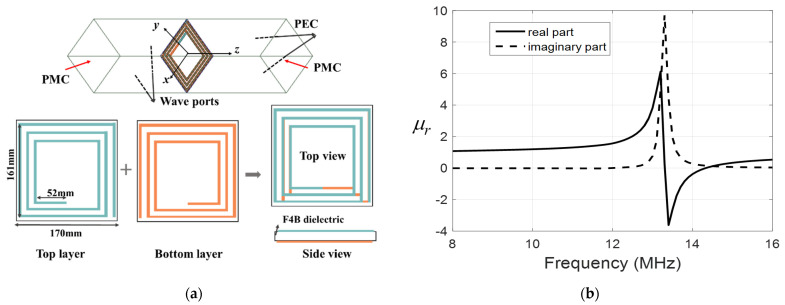
(**a**) The geometry of double−side three−turn spiral metasurface cell and periodic boundary simulation model. (**b**) The retrieval of real and imaginary parts of relative permeability of the metasurface cell.

**Figure 4 materials-15-06943-f004:**
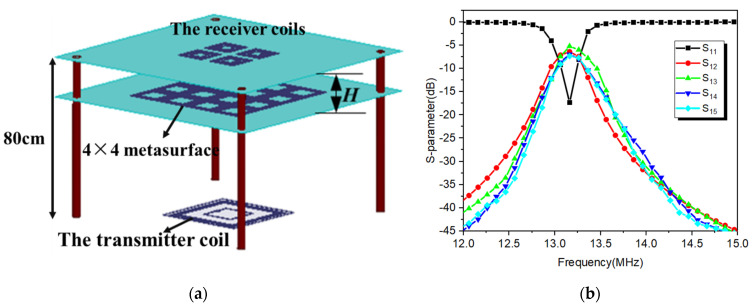
(**a**) The scenario of the proposed WPT system refers to the desk. (**b**) Reflection coefficient and transmission loss of the coils.

**Figure 5 materials-15-06943-f005:**
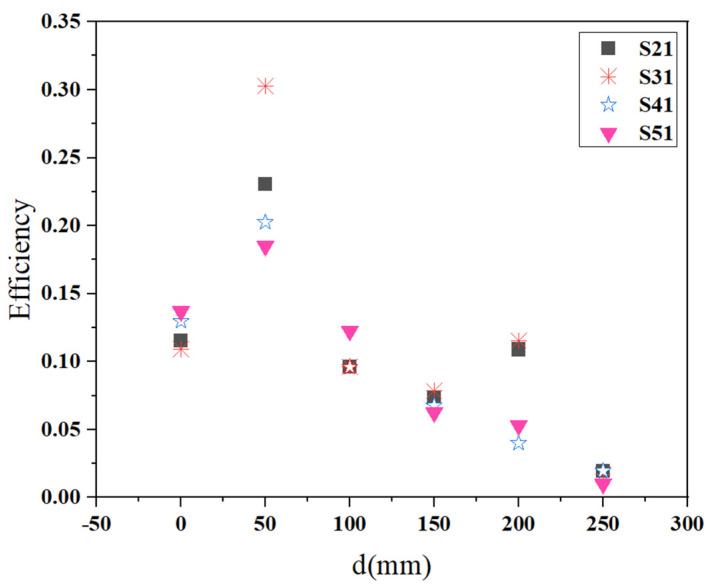
Comparison of the efficiency of the STFR system in different adjacent Rx coils.

**Figure 6 materials-15-06943-f006:**
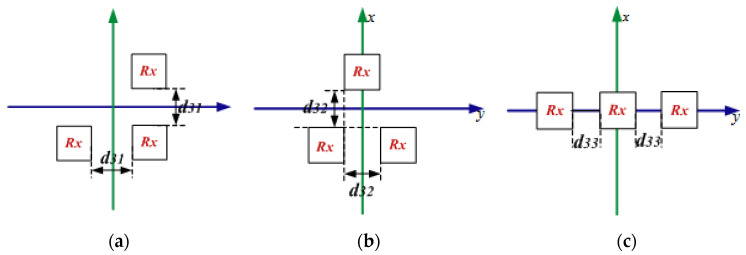
The Rx coils are in different topology positions of the STTR WPT system. (**a**) Case 0. (**b**) Case 1. and (**c**) case 2.

**Figure 7 materials-15-06943-f007:**
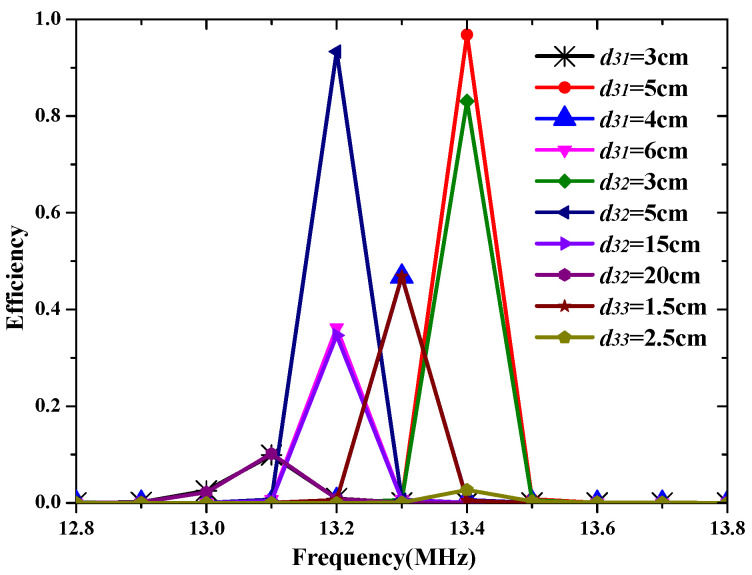
The efficiency of the STTR WPT system in a different spacing distance of the adjacent Rx coils.

**Figure 8 materials-15-06943-f008:**
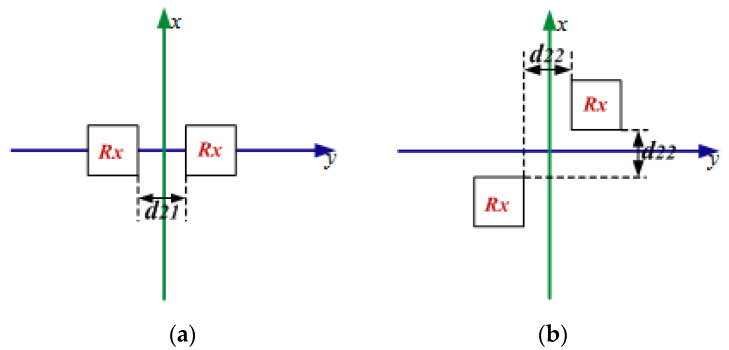
The Rx coils in different topology positions of the STTWR WPT system: (**a**) case 3 and (**b**) case 4.

**Figure 9 materials-15-06943-f009:**
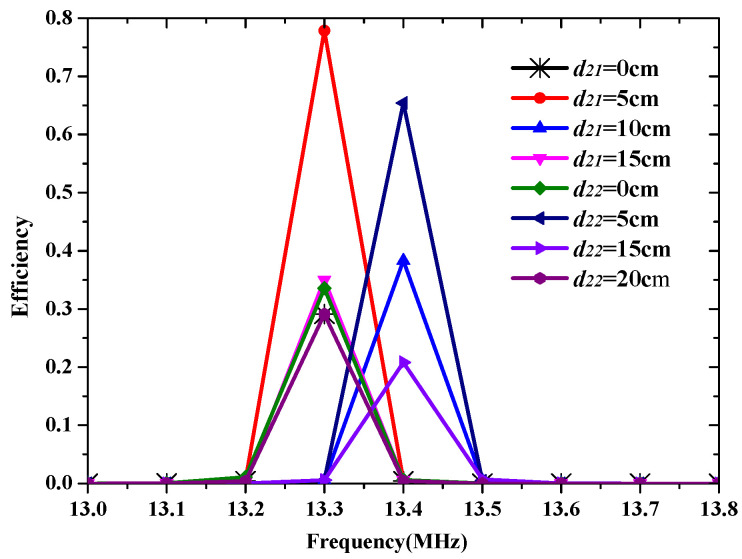
The efficiency of the STTWR/WPT system in a different spacing distance of the adjacent Rx coils.

**Figure 10 materials-15-06943-f010:**
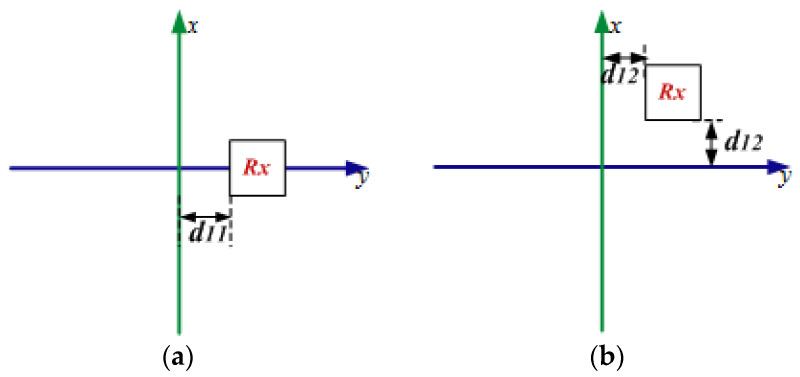
The Rx coils in different topology positions of the STSR WPT system, (**a**) case5 and (**b**) case 6.

**Figure 11 materials-15-06943-f011:**
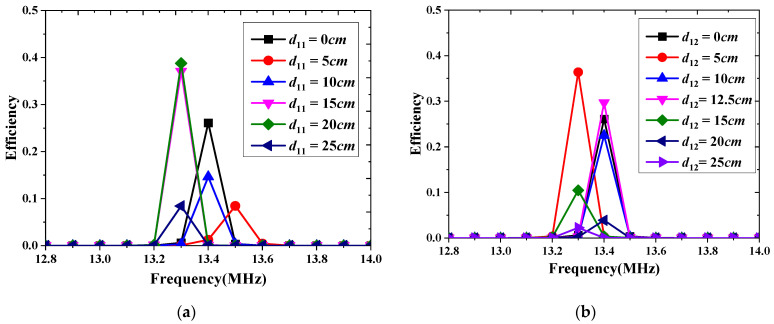
The efficiency of the STSR WPT system in different cases, (**a**) case5 and (**b**) case 6.

**Figure 12 materials-15-06943-f012:**
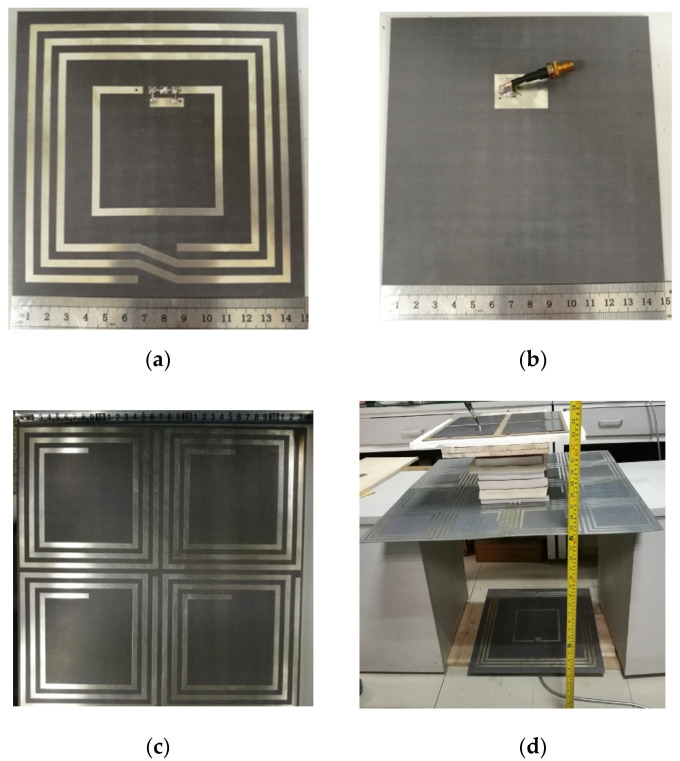
(**a**) The front view of the Tx coil. (**b**) The back view of the Tx coils. (**c**) The permeability metasurface, and (**d**) the measurement setup.

**Figure 13 materials-15-06943-f013:**
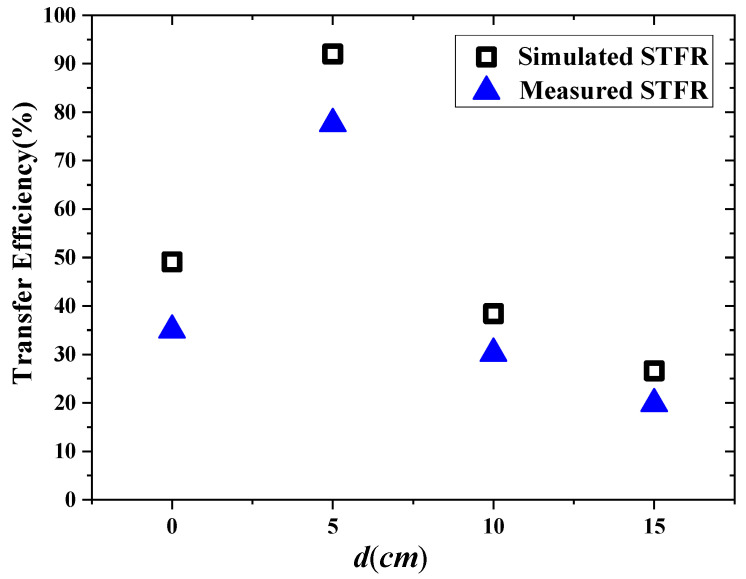
Comparison of the total efficiency of the simulation and measurement results of the STFR WPT system integrating with metasurface, respectively.

**Table 1 materials-15-06943-t001:** The parameters of the Tx and Rx coils.

Parameter	Value	Parameter	Value	Parameter	Value
s(Tx)	45	so(Tx)	40.4	se(Tx)	17.6
w(Tx)	0.8	sl(Tx)	0.6	g(Tx)	2.5
s(Rx)	15	so(Rx)	13.6	se(Rx)	6.8
w(Rx)	0.4	sl(Rx)	0.4	g(Rx)	1

Unit: cm.

**Table 2 materials-15-06943-t002:** The efficiency of conventional multiple receivers WPT system in the different distances of the adjacent Rx coils.

d(cm)	|S_21_|	|S_31_|	|S_41_|	|S_51_|	Total Efficiency
0	0.0265	0.0267	0.0273	0.0265	0.29%
5	0.0271	0.0291	0.0272	0.0274	0.31%
15	0.0235	0.0239	0.0252	0.0243	0.23%
25	0.0231	0.0224	0.0219	0.0223	0.2%

**Table 3 materials-15-06943-t003:** The efficiency of the multiple receivers WPT system integrated with metasurface in different H.

H(cm)	Frequency	|S_21_|	|S_31_|	|S_41_|	|S_51_|	Total Efficiency
10	13.7 MHz	0.16	0.16	0.17	0.16	10.6%
20	13.1 MHz	0.48	0.55	0.45	0.43	92%
30	13.2 MHz	0.28	0.3	0.27	0.24	30%
40	13.7 MHz	0.15	0.15	0.15	0.15	9%

**Table 4 materials-15-06943-t004:** The performance comparison between this work and previous work.

WPT	Tx Coils	Rx Coils	Metasurface	Resonate Frequency	Transfer Distance	Total Efficiency
Ref. [8]	30 cm (0.038λ)	30 cm (0.038λ) One Rx coil	-	8.3 MHz	3.8 cm (0.0048λ)	50%
Ref. [18]	20 cm (0.18λ)	20 cm (0.18λ) One Rx coil	μ<0	27 MHz	50 cm (0.45λ)	47%
Ref. [23]	3.1 cm (0.14λ)	0.64 cm (0.029λ) Four Rx coils	μ≈0	13.56 MHz	45 cm (2λ)	80.6%
This work	45 cm (2.1λ)	15 cm (0.7λ) Four Rx coils	μ<0	13.1 MHz	80 cm (3.62λ)	92%

## Data Availability

Data underlying the results presented in this paper are not publicly available at this time but may be obtained from the authors upon reasonable request.

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
