# Peer review of "The Efficiency Improvement of Multiple Receivers in Wireless Power Transmission by Integrating Metasurfaces"

_materials, 2022, doi:10.3390/ma15196943_

Round 1
Reviewer 1 Report
In my opinion manuscript materials-1872078 is well written and deserves publication.
Some suggestion to improve the manuscript:
1. Emphasize novelty in the abstract.
2. Emphasize originality in the Introduction
3. Revise English, especially at lines 320, 321. Also the last sentence of the section 4 seems incomplete (line 341).
4. Line 348 indicate distance for 97% efficiency.
5. In conclusion discuss the practical use of such systems.
Author Response
Responses to Reviewers’ Comments
We would like to thank the reviewers for the constructive suggestions and comments, which would help us improve the quality of the manuscript. We have revised our original manuscript carefully according to these suggestions and comments. Below are our item-to-item responses to the reviewers’ comments.
To Reviewer 1
Comment:
This letter presents efficiency improvement of multiple receivers in wireless power transmission by integrating metasurfaces. The reviewer gives comments on the following:
1. Emphasize novelty in the abstract.
Reply:
Thank you very much for your comments. According to your comments, we have emphasized the novelty of this paper in the abstract. The revisions are presented in the word manuscript.
2. Emphasize originality in the Introduction
Reply:
Thank you very much for your comments. According to your comments, we have emphasized the novelty of this paper in the introduction. The revisions are presented in the word manuscript.
3. Revise English, especially at lines 320, 321. Also the last sentence of the section 4 seems incomplete (line 341).
Reply:
Thank you very much for your comments. According to your comments, we have corrected lines 320, 321, and 341 of the original manuscript. The revisions are presented in the word manuscript.
4. Line 348 indicate distance for 97% efficiency.
Reply:
Thank you very much for your comments. In line 348, the distance for 97% efficiency is not found. The 97% efficiency belongs to the maximum efficiency of the SFFR WTP system. The manuscript is carefully revised again.
5. In conclusion discuss the practical use of such systems.
Reply:
Thank you very much for your comments. The application scenarios of the multiple receivers WPT system integrating negative permeability metasurface designed in this paper are various desks, such as desks and home desks. The Tx coil can be mounted on the floor. The metasurface is installed under the desk drawer. The Rx coils are installed on the surface of the desk to achieve multi-target charging. The revisions are presented in the word manuscript.

Reviewer 2 Report
General remarks:
Bad language. Many sentences start with ‘And’ where this is inappropriate.
Captions of tables should end with periods.
No spaces between the word ‘Table’ and the subsequent number in the text.
Specific comments:
Page 2; lines 53-54:
The remark …while all the resonance is taken from the multi-turn copper wire, which is not easy for conformation. It is not clear what is meant by this remark. How can resonance be ‘taken’ from something and in which sense is this not easy for conformation?
Page 2; line 77:
…are all suitable for conforming with the desk for convenient charging. What is meant by the desk?
Page 2; lines 77-85:
Roman numerals are used as reference to the following section, whereas the sections themselves are indicated by Arabic numbers.
Page 4; lines 149-151:
Just refer to Table 1 w.r.t coil dimensions.
Page 4; Fig. 2:
The caption does not correspond with what is depicted. (a) only shows a (transmitter?) coil and no multiple receivers can be seen. B shows the system with multiple receiver and single transmitter.
Page 5; Table 1:
S(Tx) is appears twice with a different value. I guess S(Rx) is meant on the third row.
Page 5; Eqs (6) and (7):
What is the physical meaning of the ‘S-parameter’ and its subscripts? If S indicates (complex) apparent power, the efficiency cannot be calculated like this! Also the units an Table 2 are missing. Furthermore, unless S is a unitless quantity, the efficiencies expressed in (6) and (7) do not have the same unit and can therefore not be equivalent.
Page 5; line 179:
And it can be seen that the efficiency has no evident change in different distances of d.
Albeit a very low efficiency in absolute terms is observed for all distances, a drop from 0.29% to 0.2% is still a relative drop of more than 30%! That is quite substantial.
Page 5; lines 187-196:
Very bad language throughout this section. One sentence is even repeated directly after it ended.
Page 6; lines 197:
Is effective relative permeability meant given the values on the vertical axis if Fig. 3b? Also, unit on the vertical axis is missing. Which parts in Fig. 3a are metamaterial?
Page 6; lines 199:
Then, the negative permeability metasurface is composed of 4 x 4 metasurface cells with a total dimension of 68 cm x 68 cm to optimize the position of the Metasurface where the transfer efficiency is maximum.
How were these dimensions obtained? Where is the claim of these sizes being optimal substantiated?
Page 6 and Table 3:
How are the presented results as function of H obtained? From the model described on page 3 or experimentally? In the former is the case, how does the 2D model deal with the spiral-shape? In the model the region is homogeneous and the spirals can not be modelled, let alone a finite array of these metamaterial spirals. Also, in Table 3 no units for S are shown.
Page 7; line 238:
Where is it shown that the conventional configuration is optimal. This is an unsubstantiated claim. Also, the topology is claimed to be similar to the single transmitter-four receiver topology, so it is not the same then? What was changed?
Page 8; line 245-250:
What is a possible explanation for observations made that the efficiencies are different for the 4 receiver coils for a given value of d?
Page 8; Fig. 7:
How were the distances d obtained? From an optimization using the model? Experimentally? Where they just fixed and then a frequency sweep was performed? Shouldn’t the height, dimension and configuration of the metamaterial layer be adjusted as well or were the kept fixed? If so, why? Same comments for section 3.3 and 3.4
Page 10; lines 324-329:
One of the factors causing discrepancy between simulation and experiment is claimed to be the difficulty w.r.t. positioning. This can easily be solved and is in my opinion not a valid argument when the set-up is properly designed. Judging from fig. 12 a lot of improvement can be made in this area. Another factor is the difference in permeability. Then why not first try to measure/characterize it. The discrepancy on account of 2D model assumptions is not addressed at all.
Page 11; lines 324-329:
Why, make this comparison with systems in literature? Can these be compared honestly? We know nothing of these systems in terms of dimensions, configuration, power rating etc., based on the text of the paper only.
Author Response
Responses to Reviewers’ Comments
We would like to thank the reviewers for the constructive suggestions and comments, which would help us improve the quality of the manuscript. We have revised our original manuscript carefully according to these suggestions and comments. Below are our item-to-item responses to the reviewers’ comments.
To Reviewer 2
Comment:
This letter presents efficiency improvement of multiple receivers in wireless power transmission by integrating metasurfaces. The reviewer gives comments on the following:
1. Page 2; lines 53-54:
The remark …while all the resonance is taken from the multi-turn copper wire, which is not easy for conformation. It is not clear what is meant by this remark. How can resonance be ‘taken’ from something and in which sense is this not easy for conformation?
Reply:
Thank you very much for your comments. The reference [11] introduced on lines 53-54 were replaced by the reference that more focused on the work of this paper. Therefore, we deleted lines 53-54 of the original manuscript. The introduction has been further modified.
2. Page 2; line 77:
…are all suitable for conforming with the desk for convenient charging. What is meant by the desk?
Reply:
Thank you very much for your comments. We have modified Line 77 to change 'conforming' to 'conformal'. The Tx coil is installed on the floor, the Rx coil is installed on the surface of the desk, and the metasurface is installed on the lower surface of the desk drawer. The design of this work can be conformal to the desk.
3. Page 2; lines 77-85:
Roman numerals are used as reference to the following section, whereas the sections themselves are indicated by Arabic numbers.
Reply:
Thank you very much for your comments. We changed the Roman numerals for lines 77-85 to Arabic numerals.
4. Page 4; lines 149-151:
Just refer to Table 1 w.r.t coil dimensions.
Reply:
Thank you very much for your comments. Lines 149-151 are replaced by the sentence ‘The geometrical parameters of the Tx and Rx coils are shown in Table1’.
5. Page 4; Fig. 2:
The caption does not correspond with what is depicted. (a) only shows a (transmitter?) coil and no multiple receivers can be seen. B shows the system with multiple receiver and single transmitter.
Reply:
Thank you very much for your comments. We have modified the caption of Figure 2(a) so that the caption corresponds to the picture.
6. Page 5; Table 1:
S(Tx) is appears twice with a different value. I guess S(Rx) is meant on the third row.
Reply:
Thank you very much for your comments. We changed S(Tx) in the third row of table 1 to R(Tx).
7. Page 5; Eqs (6) and (7):
What is the physical meaning of the ‘S-parameter’ and its subscripts? If S indicates (complex) apparent power, the efficiency cannot be calculated like this! Also the units an Table 2 are missing. Furthermore, unless S is a unitless quantity, the efficiencies expressed in (6) and (7) do not have the same unit and can therefore not be equivalent.
Reply:
Thank you very much for your comments. The Tx coil is port 1 and the Rx coils are ports 2, 3, 4, and 5. The S-parameter is used to reflect the relationship between the incident voltage wave and the reflected voltage wave at the port. S11 is the reflection coefficient of port 1 when the other ports are matched. S21, S31, S41, and S51 are the transmission coefficients from port 1 to ports 2, 3, 4, and 5 when ports 2, 3, 4, and 5 are matched. The S11, S21, S31, S41, and S51 are complex. |S11|, |S21|, |S31|, |S41|, and |S51| are real and used to calculate the efficiency. The S11, S21, S31, S41, and S51 are modified to |S11|, |S21|, |S31|, |S41| and |S51| in Table2. The S-parameter has no unit and is a ratio. Therefore, the efficiency expressed in (6) and (7) can be equivalent.
8. Page 5; line 179:
And it can be seen that the efficiency has no evident change in different distances of d.
Albeit a very low efficiency in absolute terms is observed for all distances, a drop from 0.29% to 0.2% is still a relative drop of more than 30%! That is quite substantial.
Reply:
Thank you very much for your comments. We have revised line 179 to make the expression more correct without ambiguity. Line 179 was modified to ‘It can be seen that the efficiency still cannot be improved to more than 31% under different d’.
9. Page 5; lines 187-196:
Very bad language throughout this section. One sentence is even repeated directly after it ended.
Reply:
Thank you very much for your comments. We have revised this important issue raised by the reviewer. Sorry for such an error due to our negligence. Thanks again to the reviewers for their serious, careful, and rigorous review of this work.
10 .Page 6; lines 197:
Is effective relative permeability meant given the values on the vertical axis if Fig. 3b? Also, unit on the vertical axis is missing. Which parts in Fig. 3a are metamaterial?
Reply:
Thank you very much for your comments. Figure. 3(b) is the relative permeability which is no unit. We modified the expression of permeability in line 197 to be relative permeability and modified the vertical axis of Figure. 3(b) to be relative permeability .
11. Page 6; lines 199:
Then, the negative permeability metasurface is composed of 4 x 4 metasurface cells with a total dimension of 68 cm x 68 cm to optimize the position of the Metasurface where the transfer efficiency is maximum. How were these dimensions obtained? Where is the claim of these sizes being optimal substantiated?
Reply:
Thank you very much for your comments. The metasurface size of 4×4 is based on the size of the Tx coil (45cm) and the Rx coils (15cm×2). The metasurface is larger than the Tx and Rx coils so that the magnetic field emitted by the Tx coil can be completely converged on the Rx coil, thereby obtaining greater efficiency. If the metasurface is too small, the improvement of efficiency is small. if it is too large, it is not suitable for practical applications. The magnetic distribution of the multiple receivers WPT system under metasurfaces of different sizes is shown in Figure. 1. It can be seen that the magnetic field at the Rx coils is weaker when the metasurface size is 3 × 3. As the size of the metasurface increases from 3 × 3 to 5 × 5, the magnetic field at the Rx coils first increases and then decreases. Therefore, increasing the size of the metasurface can increase the magnetic field strength at the Rx coils. Thereby, the transfer efficiency of the system is improved. However, if the metasurface is too large, the magnetic field at the Rx coils will be weakened, so the size of the selected metasurface is generally larger than that of the Tx coil. Therefore, the size of the metasurface selected in this work is 4 × 4, and the magnetic field at the Rx coils is the largest at this time. The optimal mentioned in line 199 refers to the optimal position of the metasurface between the Tx coil and the Rx coils. This position is obtained by the optimization process in Table 3.
12. Page 6 and Table 3:
How are the presented results as function of H obtained? From the model described on page 3 or experimentally? In the former is the case, how does the 2D model deal with the spiral-shape? In the model the region is homogeneous and the spirals can not be modelled, let alone a finite array of these metamaterial spirals. Also, in Table 3 no units for S are shown.
Reply:
Thank you very much for your comments. In a multiple receivers WPT system integrating with metasurface, it is difficult to express the relationship between the distance H and the transfer efficiency by a certain functional relationship. There is no similar work in the currently known literature. Therefore, the effect of the distance H on the transfer efficiency can only be obtained by optimizing, as shown in Table 3. The helix in this paper is a planar helical structure as shown in Figure.2. The two spiral structure is printed on the upper and lower layers of the 1 mm thickness F4B dielectric substrate, respectively. The upper and lower spiral structures are mirror symmetric about the center of the structure. The top view of the 4×4 metasurface is shown in Figure. 3. The S-parameter is no unit and is modified in Table 3.

Reviewer 3 Report
1. In the article, symbols are used in the formulas, which are written in italics. The same symbols are used in the text and should also be in italics (e.g. on lines 111, 114 and in Figure 1). Necessary to unify the text.
2. Figure 2b is hardly understandable. What does "z" mean, is it an axis? If so, this line should be extended so as not to confuse the reader. From the present drawing it can be read that this is some distance and that the Transmitter and Receivers system is shifted.
3. With a larger distance between the coils, the efficiency should decrease. So why is the efficiency greater for d = 5 (Table 2)?
4. What was the reason why F4B was chosen? What were the advantages and disadvantages of other solutions? This information should be added to the article. did the authors compare the results with other surfaces?
5. Figure 3 is far too large for the text of the article.
6. Why is the efficiency so low at H = 10 and amounts to only 10.6% (Table 3)? Why were there no results for other values ​​(among those in the table)? I believe that there should also be other distances to even show the relationship (Table 2, 4).
7. Introduction, according to my opinion, includes a small literature review. The Introduction should also include issues covered in the publications:
a) Wang, B .; Yerazunis, W.; Teo, K.H. Wireless Power Transfer: Metamaterials and array of coupled resonators. Proc. IEEE 2013, 101, 1359–1368.
b) Stankiewicz, J.M .; Choroszucho, A. Comparison of the Efficiency and Load Power in Periodic Wireless Power Transfer Sys-tems with Circular and Square Planar Coils. Energies 2021, 14, 4975.
c) Christ, A .; Douglas, M.G .; Roman, J.M .; Cooper, E.B .; Sample, A.P .; Waters, B.H .; Kuster, N. Evaluation of wireless resonant power transfer systems with human electromagnetic exposure limits. IEEE Trans. Electromagn. Compat. 2013, 55, 265–274.
8. There should no longer be a drawing and a table in the Conclusions section. This should have been added to the previous chapter. In Conclusions, please write down the advantages and disadvantages of the proposed solution in relation to others, provide opportunities for developing the proposed solution.
9. I believe that the article in the Introduction section should be more enriched by highlighting the solutions used in other publications, highlighting the advantages and disadvantages of the proposed solutions.
10. The article presents experimental results. Have they been verified numerically or analytically? On what basis was the correctness of the results assumed?
11. In what real cases can the proposed solution be used?
Author Response
Responses to Reviewers’ Comments
We would like to thank the reviewers for the constructive suggestions and comments, which would help us improve the quality of the manuscript. We have revised our original manuscript carefully according to these suggestions and comments. Below are our item-to-item responses to the reviewers’ comments.
To Reviewer 3
Comment:
This letter presents efficiency improvement of multiple receivers in wireless power transmission by integrating metasurfaces. The reviewer gives comments on the following:
1. In the article, symbols are used in the formulas, which are written in italics. The same symbols are used in the text and should also be in italics (e.g. on lines 111, 114 and in Figure 1). Necessary to unify the text.
Reply:
Thank you for your comments. The symbols are used in the text and the formulas have been changed to italics the Section 2. The revisions are presented in the word manuscript.
2. Figure 2b is hardly understandable. What does "z" mean, is it an axis? If so, this line should be extended so as not to confuse the reader. From the present drawing it can be read that this is some distance and that the Transmitter and Receivers system is shifted.
Reply:
Thank you for your comments. ‘z’ is indeed the axis. And this line has been extended so as not to confuse the reader. Figure 2b is a 3D view, so it can lead to some misunderstandings when the z-axis doesn't go beyond the bounds of the model. And we have updated Figure 2 in the manuscript.
3. With a larger distance between the coils, the efficiency should decrease. So why is the efficiency greater for d = 5 (Table 2)?
Reply:
Thank you for your comments. In [1], a key issue for powering of multiple receivers is the coupled mode frequency splitting that occurs when two receivers are in close enough proximity that their magnetic fields are relatively strongly coupled. This makes the multiple receivers WPT system has worse total efficiency when the mutual coupling of the Rx coils is strong. When the distance between the Rx coils is less than 5 cm, the coupling between the Rx coils is too strong, resulting in frequency splitting. As a result, the efficiency of the target frequency is reduced. When the distance between the Rx coils is greater than 5 cm, the coupling between the Rx coils becomes smaller. But the aperture efficiency becomes lower. Therefore, the transfer efficiency is also reduced. When the distance between the Rx coils is 5 cm, it is the optimal position for the maximum efficiency after the coupling and aperture efficiency are balanced.
[1] Cannon B L , Hoburg J F , Stancil D D , et al. Magnetic Resonant Coupling As a Potential Means for Wireless Power Transfer to Multiple Small Receivers[J]. IEEE Transactions on Power Electronics, 2009, 24(7).
4. What was the reason why F4B was chosen? What were the advantages and disadvantages of other solutions? This information should be added to the article. did the authors compare the results with other surfaces?
Reply:
Thank you for your comments. The commonly used dielectric substrate in the industry is Rogers, F4B, and FR4. Among them, the high cost of Rogers dielectric is not suitable for widespread promotion. The remaining two dielectrics are cheap. Among them, the loss tangent of F4B is 0.001, and the loss tangent of the FR4 sheet is 0.025. Therefore, FR4 has a large loss of electromagnetic waves, which will affect the transfer efficiency of the WPT system. So, this work chooses F4B. The advantages of F4B are added to the manuscript. Currently, there is almost no difference in transfer efficiency between the coils we machined with Rogers dielectric and the coils machined with F4B. In addition, we use the FR4 dielectric in the microstrip antenna, and it can be found that FR4 has a significant impact on the efficiency of the antenna, and the efficiency becomes lower.
5. Figure 3 is far too large for the text of the article.
Reply:
Thank you for your comments. The size of Figure 3 has been adjusted in the manuscript.
6. Why is the efficiency so low at H = 10 and amounts to only 10.6% (Table 3)? Why were there no results for other values (among those in the table)? I believe that there should also be other distances to even show the relationship (Table 2, 4).
Reply:
Thank you for your comments. The H is the distance between the metasurface and Rx coils. When the metasurface is too close to the Rx coils, the coupling between the Rx coils increases. This results in a split in the resonant frequency, thus resulting in a significant reduction in efficiency at the resonant frequency. Therefore, the efficiency is only 10.6% when H is 10 cm. In addition, the distance between the metasurface and Rx coils has an important influence on the transfer efficiency. The three cases are given to illustrate the effect of metasurface position on the transfer efficiency. When the metasurface is closer to the Rx coils, the incident angle θ1 at which the magnetic field radiated from the Tx coil reaches the metasurface becomes larger due to the divergent properties of electromagnetic waves. According to the formula given in Figure. 1, the focusing angle θ2 also increases accordingly. Therefore, part of the magnetic field is not received by the Rx coils at the edge of the Rx coils as shown in Figure. 1(b), resulting in a decrease in the efficiency of the WPT system. When the metasurface is closer to the Tx coil, the magnetic field is bunched before it diverges, resulting in a long distance from the metasurface to the Rx coils after bunching. Therefore, the magnetic field is not all received by the Tx coil due to the divergence, as shown in Figure. 1(c). So, the transfer efficiency is reduced. The optimal position of the metasurface is shown in Figure. 1(a) which the magnetic field is all received by the Tx coil. In this case, the metasurface focuses all the magnetic fields emitted by the Tx coil and are fully received by the Rx coil, as shown in Figure. 1(a). This greatly improves the transfer efficiency. Therefore, the position of the metasurface is needed to be optimized that can obtain maximum transfer efficiency. And the position of the metasurface must not be very close to the Tx and Rx coils.
Figure. 1 (a) The metasurface is in the mid-position. (b) The metasurface is closer to the Rx coil. (c) The metasurface is closer to the Tx coil.
Table 3 uses the optimal spacing of the Rx coils in Table 2 to analyze the influence of the Rx coils and the metasurface distance on the transfer efficiency. If the distance between the Rx coils and the metasurface is analyzed using the other Rx coils spacings in Table 2, the same trend as those in Table 3 will be obtained. Therefore, the influence of the distance between the Rx coils and the metasurface on the transfer efficiency under other Rx coils spacings is not given in the paper. The efficiency curves of the multiple receivers WPT system loaded with metasurfaces under different H is shown in Figure. 2. It can be seen that the metasurface is most efficient when the distance from the Rx coils is H=20cm.
Figure. 2 The efficiency of multiple receivers with metasurfaces at different H.
7. Introduction, according to my opinion, includes a small literature review. The Introduction should also include issues covered in the publications:
a) Wang, B .; Yerazunis, W.; Teo, K.H. Wireless Power Transfer: Metamaterials and array of coupled resonators. Proc. IEEE 2013, 101, 1359–1368.
b) Stankiewicz, J.M .; Choroszucho, A. Comparison of the Efficiency and Load Power in Periodic Wireless Power Transfer Systems with Circular and Square Planar Coils. Energies 2021, 14, 4975.
c) Puccetti, G .; Stevens, C.J .; Reggiani, U .; Sandrolini, L. Experimental and numerical investigation of termination impedance effects in wireless power transfer via metamaterial. Energies 2015, 8, 1882–1895.
d) Christ, A .; Douglas, M.G .; Roman, J.M .; Cooper, E.B .; Sample, A.P .; Waters, B.H .; Kuster, N. Evaluation of wireless resonant power transfer systems with human electromagnetic exposure limits. IEEE Trans. Electromagn. Compat. 2013, 55, 265–274.
Reply:
Thank you for your comments. According to your comments, we present the relevant published research work in the introduction to highlight the significance of this work. The revisions are presented in the word manuscript.
8. There should no longer be a drawing and a table in the Conclusions section. This should have been added to the previous chapter. In Conclusions, please write down the advantages and disadvantages of the proposed solution in relation to others, provide opportunities for developing the proposed solution.
Reply:
Thank you for your comments. The pictures and tables are removed from the conclusion section and are added to the previous chapter. The conclusion is enriched that the advantages and disadvantages of the proposed solution in relation to others. The revisions are presented in the word manuscript.
9. I believe that the article in the Introduction section should be more enriched by highlighting the solutions used in other publications, highlighting the advantages and disadvantages of the proposed solutions.
Reply:
Thank you very much for your comments. According to your comments, we have enriched the introduction by highlighting the solutions used in other publications. And the advantages and disadvantages of the proposed solutions are also given. The revisions are presented in the word manuscript.
10 The article presents experimental results. Have they been verified numerically or analytically? On what basis was the correctness of the results assumed?
Reply:
Thank you for your comments. The experimental results of this work are measured by the vector network analyzer. By the measurement, the S-parameter of Tx and Rx coils can be obtained, such as S12, S13, etc. This test method is widely used in WPT system efficiency testing, and many previous works have verified the correctness of this method through simulation and experiments. Then, the efficiency of Rx coils in multiple receivers WPT system can be calculated as [1]:
Therefore, the experimental results obtained in this paper are also reasonable by using mature test methods and correct efficiency calculation methods that have been proved in [1].
[1] Cannon B L, Hoburg J F, Stancil D D, et al. Magnetic Resonant Coupling As a Potential Means for Wireless Power Transfer to Multiple Small Receivers. IEEE Transactions on Power Electronics 2009, 24(7), 1819-1825.
11. In what real cases can the proposed solution be used?
Reply:
Thank you for your comments. The application scenarios of the multiple receivers WPT system integrating negative permeability metasurface designed in this paper are various desks, such as desks and home desks. The Tx coil can be mounted on the floor. The metasurface is installed under the desk drawer. The Rx coils are installed on the surface of the table to achieve multi-target charging. The specifics of the revision are presented in the word manuscript through revision.

Reviewer 4 Report
There will be some results i want to know from the author
1. please provide s11 return loss result together with s21 s32… to showmit matches.
2. could you add some magnetic coupling distribution
3. the presence of the metamaterial is just like repeater. Could you provide the distance between the metamaterial with the transmitter and receiver. Compare the result with the repeater.
Author Response
Responses to Reviewers’ Comments
We would like to thank the reviewers for the constructive suggestions and comments, which would help us improve the quality of the manuscript. We have revised our original manuscript carefully according to these suggestions and comments. Below are our item-to-item responses to the reviewers’ comments.
To Reviewer 4
Comment:
This letter presents ‘Efficiency Improvement of Multiple Receivers in Wireless Power Transmission by Integrating Metasurfaces’. The reviewer gives comments in the following:
1. Please provide s11 return loss result together with s21 s32… to show it matches.
Reply:
Thank you very much for your comments. Figure 1(a) shows the proposed multiple receivers WPT system loaded with metasurfaces, where the distance H=20 cm from the metasurface to the Rx coils. Maximum efficiency can be obtained at this distance. The S-parameters of the Tx coil and Rx coils in the system is shown in Figure. 1(b). It can be seen that the 1 port of the Tx coil is matched at 13.1MHz, and the transmission coefficients from the Tx coil to the Rx coils are about -6dB.
Figure. 1 (a) The geometry of the proposed multiple receivers WPT system. (b) S-parameter of the proposed multiple receivers WPT system.
2. Could you add some magnetic coupling distribution
Reply:
Thank you very much for your comments. The magnetic field distribution of the multiple receivers WPT system after loading the metasurface is shown in Figure. 2. It can be seen that when the distance from the metasurface to the Rx coils is H=20cm, the magnetic field at the Rx coils is the largest. When H is less than or greater than 20 cm, the magnetic field at the Rx coils becomes smaller. Therefore, when the distance H between the metasurface and the Rx coils is 20 cm, the transfer efficiency is the highest. In addition, Figure. 2 also shows the magnetic field distribution of the multiple receivers WPT system when the metasurface is not loaded. It can be seen that the magnetic field at the Rx coils is very weak when the metasurface is not loaded. Therefore, the introduction of metasurfaces in WPT systems can greatly improve transfer efficiency.
Figure. 2 The magnetic distribution of the multiple receivers WPT system.
3. Could you provide the distance between the metamaterial with the transmitter and receiver. Compare the result with the repeater.
Reply:
Thank you very much for your comments. The metasurfaces in the multiple receivers WPT system loaded with metasurfaces are replaced with the relay coil, where the relay coil and the Tx coil are the same sizes. The distance from the relay coil to the Rx coils is also H. Then, the efficiency curves of the multiple receivers WPT system loaded with metasurfaces and the multiple receivers WPT system loaded with relay coils under different H are obtained, as shown in Figure. 3. It can be seen that the metasurface is most efficient when the distance from the Rx coils is H=20cm. The relay coil has the highest efficiency when the distance from the Rx coils is H=40cm, which is located in the middle of the Rx coils and the Tx coil. Furthermore, the maximum efficiency of the multiple receivers WPT system after the introduction of the metasurface is greater than that of the introduction of the relay coil. This is because metasurfaces have a stronger ability to focus magnetic fields than relay coils.
Figure. 3 The efficiency of multiple receivers with metasurfaces at different H and the efficiency of multiple receivers with the relay coil at different locations

Round 2
Reviewer 2 Report
The authors have attempted to make adjustments to their paper in response to my review to the best of their ability . Some clarifications were only made in the response letter to the remarks I had after the first review. I expect some of this information to be included in the paper and not in a personal correspondence, since the questions I raised other readers may have as well. This can be improved still. However, I have some more fundamental objections in the manner in which this research has been conducted, judging from how the scientific work is presented in the paper.
It is not clear how the figures of section 3 were obtained. I expect them to be the result of simulations. What is unclear, is which model was used to create them. Was it by the 1D analytical model of equations 1 and 2 and figure 1? If so, how come that a relative good correspondence is found between experiment and simulations for a configuration that is clearly 3D in nature, given the rectangular shape of the coils and their spatial configuration. It is mentioned in the text that a FEM (ANSYS) model was used ‘to simulate the Tx and Rx coils’. Nowhere in the text is clearly explained how the result of the graphs were obtained. Neither is it explained what the role/function of the two models is and what the impact is of the assumptions made in the 1D analytical model. The same holds for the results presented in tables
The experimental verification is flawed in my opinion for the following reasons:
1. The set-up is substandard, because the distance between Tx and the metasurface is create by a pile of books (!!!)… That is an unscientific and ill-defined way to create a clearance/distance to verify the theoretical models.
2. It is stated that the electromagnet properties of the desk and books were unknown and therefore are the root cause of discrepancies. For model verification a ‘clean’ set-up that resembles what was simulated as closely as possible can be easily obtained. The meta surface and receivers coils can be suspended in air or be placed on surfaces of nonconductive, nonferrous material. The uncertainty of the desk and books is then eliminated altogether. The experiment on the presented set-up can still be conducted to quantify the effect of the desk and books to mimic a more realistic situation by comparing them to the ‘clean’ set-up.
All in all, these concerns make it very hard to reproduce the work presented in the paper and the claims made cannot be easily verified.
Author Response
Responses to Reviewers’ Comments
We would like to thank the reviewers for the constructive suggestions and comments, which would help us improve the quality of the manuscript. We have revised our original manuscript carefully according to these suggestions and comments. Below are our item-to-item responses to the reviewers’ comments.
To Reviewer 2
Comment:
This letter presents ‘Efficiency Improvement of Multiple Receivers in Wireless Power Transmission by Integrating Metasurfaces’. The reviewer gives comments in the following:
- I expect some of this information to be included in the paper and not in a personal correspondence
Reply:
Thank you very much for your comments. The comments mentioned in the first review have been added in the revised manuscript. The revised part is shown in the red part of the paper. Thank you very much again.
- It is not clear how the figures of section 3 were obtained. I expect them to be the result of simulations. What is unclear, is which model was used to create them. Was it by the 1D analytical model of equations 1 and 2 and figure 1? If so, how come that a relative good correspondence is found between experiment and simulations for a configuration that is clearly 3D in nature, given the rectangular shape of the coils and their spatial configuration.
Reply:
Thank you very much for your comments. The description of Figure 2 is as follows:
Both the transmit and receive coils in this paper are 2D models. That is, the coil is printed on the upper surface of the 2mm thick F4B dielectric. Figure 2(a) is a top view of the Tx coil. In order to better show the structure of the Tx coil, a side view of the Tx coil is added to Figure 2(b). The Rx coil has the same structure as the Tx coil, but the size of the Rx coil is smaller than that of the Tx coil. Figure 2(c) is a diagram of a transceiver system composed of a Tx coil and four Rx coils. The introduction mentioned above has been updated in the corresponding place in the paper
The description of Figure 3 is as follows: Figure 3(a) shows the detailed geometry of the metasurface cell and its simulation model in HFSS (High Frequency Simulation Software). The metasurface cell adopts the periodic boundary simulation method. The two planes perpendicular to the z-axis are set as the magnetic boundary (PMC). The two faces perpendicular to the x-axis are set as electrical boundaries (PEC). The two planes perpendicular to the y-axis are set as the wave ports. And the purpose of this setting is to let the magnetic field pass perpendicularly through the metasurface cell, as shown in Figure. 3(a). The two-spiral structure of the metasurface cell is printed on the upper and lower layers of the 1 mm thickness F4B dielectric substrate, respectively. The relative permeability curve in Figure. 3(b) is obtained by simulating the metasurface cell in HFSS and using the Smith inversion algorithm.
The description of Figure 4 is as follows: Figure 4(a) is the multi-receiver WPT model after integrating 4 × 4 metasurfaces. The model is simulated in HFSS to extract SNP files into ADS for port matching of Tx and Rx coils. The LC matching circuit is obtained, and then the S-parameters of the system are obtained by simulation and further optimization in HFSS, as shown in Figure. 4(b). It can be seen that the 1 port of the Tx coil is matched at 13.1MHz, and the transmission coefficients from the Tx coil to the Rx coils are about -6dB.
The other simulation results in the section 3 are matched by bringing the SNP files obtained by HFSS simulation into ADS. Then the matching circuit is brought into HFSS simulation and optimized. The related description mentioned above has been added in the revised manuscript.
- It is mentioned in the text that a FEM (ANSYS) model was used ‘to simulate the Tx and Rx coils’. Nowhere in the text is clearly explained how the result of the graphs were obtained. Neither is it explained what the role/function of the two models is and what the impact is of the assumptions made in the 1D analytical model. The same holds for the results presented in tables
Reply:
Thank you very much for your comments. The S-parameters were obtained by simulating the metasurface cell with wave ports and periodic boundary conditions. Then the NRW method is used to obtain the relative permeability as shown in Figure. 3(b). The other figures and tables involved in the simulation results in this paper are to import the S5P file obtained in the HFSS simulation into ADS to match each port. Then the matched LC circuit is brought into HFSS for further optimization. The section 2 is mainly to introduce the principle that introducing a negative permeability metasurface between the transceiver coils can improve the transmission efficiency. The model of Figure. 1 has nothing to do with the section 3, it is just a schematic diagram. Furthermore, the results of the section 3 cannot be obtained from the section 2. The section 2 is only a qualitative analysis, and the qualitative analysis can only be obtained through the simulation software used in the section 3. The introduction mentioned above has been updated in the corresponding place in the paper
- The experimental verification is flawed in my opinion for the following reasons.
Reply:
Thank you very much for your comments. The measurement in the paper is built considering the actual application scenario. The proposed multi-receiver WPT system application scenario is a desk. So, the metasurface is placed on the desk surface, with a stack of books separated between the metasurface and the Rx coils. It is used to simulate the situations in the usual desk drawer. In addition, the transfer of energy between the magnetic resonance coupling coils is achieved by a magnetic field. Books are dielectrics with a permittivity of 2.5. Therefore, the impact on the multi-receiver WPT system is small. Then, the influence of books on metasurfaces is analyzed. Suppose the books are a dielectric block of dimensions ds×ds×H with a dielectric constant of 2.5, as shown in Fig. 1(a). Then the effect of books on the magnetic permeability of the metasurface cell is analyzed, as shown in Fig. 1(b). It can be seen that the larger the book, the smaller the effect on the magnetic permeability of the metasurface cell. The size of the books is generally greater than ds = 160 mm. Therefore, it can be seen from the analysis that the books have little effect on the properties of the metasurface. The measurement efficiency and simulation efficiency of the WPT system in this paper are 77.6% and 92%, respectively. The general measurement and simulation efficiency error are reasonable within 10%. The difference between our measurement and simulation efficiency is 14.4%. This is because our measurement considers the errors brought by the actual application scenario, which is more valuable for reference.

Reviewer 3 Report
Thanks for the answers. I have no more questions.
Author Response
Thank you very much for your comments.
Reviewer 4 Report
There is no S11 result (return loss). Could you provide the result as i asked in the previous review?
Author Response
Responses to Reviewers’ Comments
We would like to thank the reviewers for the constructive suggestions and comments, which would help us improve the quality of the manuscript. We have revised our original manuscript carefully according to these suggestions and comments. Below are our item-to-item responses to the reviewers’ comments.
To Reviewer 4
Comment:
This letter presents ‘Efficiency Improvement of Multiple Receivers in Wireless Power Transmission by Integrating Metasurfaces’. The reviewer gives comments in the following:
- Please provide s11 return loss result together with s21 s32… to show it matches.
Reply:
Thank you very much for your comments. Maximum efficiency can be obtained when the distance from the metasurface to the Rx coils is H = 20 cm. The S-parameters of the Tx coil and Rx coils in this distance is shown in the following Figure. It can be seen that the 1 port of the Tx coil is matched at 13.1MHz, and the transmission coefficients from the Tx coil to the Rx coils are about -6dB. The following figure has been added in the revised manuscript.
